# Tuberculosis commodities supply chain performance in the WHO African region: A scoping review

**Alison T. Mhazo**[1]*, **Stanford Miyango**[1], **Lifton Palani**[1], **Charles C. Maponga**[2]

**1** Ministry of Health Malawi, Community Health Sciences Unit, National TB & Leprosy Elimination Program [NTLEP], Lilongwe, Malawi, **2** Department of Pharmacy and Pharmaceutical Sciences, Faculty of Medicine and Health Sciences, University of Zimbabwe, Harare, Zimbabwe

* alisonmhazo@gmail.com

**Data Availability Statement:** Since it is a review, relevant data sources are publicly available. All relevant data are contained within the manuscript.

**Funding:** The authors received no specific funding for this work.

## Abstract

Procurement and supply chain management [PSM] systems remain a critical pillar for the implementation of Directly Observed Therapy [DOTS] for tuberculosis [TB] and achievement of disease related aspirations such as 'ending TB by 2030'. We conducted a scoping review of literature using the Arksey and O'Malley [2005] framework to summarize and disseminate the results of available evidence in relation to TB commodities supply chain performance in the WHO African Region. We searched three electronic databases complemented by google search using relevant terms and identified 1,586 sources. Twenty-five studies published between 2009 and 2023 met the eligibility criteria, inclusive of 21 peer reviewed publications and four reports. The strengths we found included the existence of pooled procurement mechanisms [PPM], availability of funding through external sources, existence of logistics management information systems [LMIS] and integration of PSM systems into primary health care. The main challenge was frequent stock outs which mainly affected medicines for treating TB in children and those for preventing TB. Stock outs were found to follow a poverty gradient and pervasively inequitable since they disproportionately affected structurally disadvantaged populations and communities. Countries that rely on domestic mechanisms for procurement tend to be more vulnerable to stock outs due to inadequate and unpredictable financing, delayed disbursements of funds, longer procurement lead times and poor supplier management. We concluded that stock outs remain one of the foremost challenges to ending TB by 2030. We recommend leveraging existing performance-enhancing platforms such as PPMs, including utilization of such mechanisms by countries that utilize domestic resources to procure commodities. We recommend the design of people centric supply chains that are sensitive to the differentiated needs of the population to ensure that composite improvements in PSM performance do not mask underlying disparities. Context-relevant research is needed to inform future strategies for improving PSM performance.

**Competing interests:** The authors have declared that no competing interests exist.

## Introduction

Despite that it is preventable and curable, in 2022, Tuberculosis (TB) was the world's second leading cause of death from a single infectious agent after coronavirus disease (COVID-19), and caused almost twice as many deaths as HIV/AIDS [1]. In 1993, the World Health Organization (WHO) declared tuberculosis a global emergency and began promoting a management strategy called directly observed therapy short course [(DOTS) [2]. DOTS has five key components as identified by WHO, including a regular and uninterrupted supply of all essential anti-tuberculosis drugs. Although the initial framing on DOTS was centered on health worker supervision for TB chemotherapy, more patient centered models of TB care [3] such as family member and community based DOTS have also emerged as effective strategies in TB control [4]. Since its launch, DOTS and its emergent variants have remained a cornerstone for controlling TB globally [5] and it has informed successive frameworks and targets for international and national efforts aimed at controlling the disease [6, 7]. Since the 2000s, there had been a number of declarative commitments for TB that rely on the introduction of new tools [(diagnostics and medicines)]. As a result, PSM for TB has risen to the agenda, and it is singled out as one of the priority actions under the contemporary Global Plan to End TB [2023–2030]; a plan for ending tuberculosis [(TB)] as a public health challenge by 2030 [8].

Despite the centrality of TB commodities in fulfilling the effective implementation of DOTS, there is limited knowledge on how the performance of supply chains affects the availability of those commodities, and ultimately achievement of declarative aspirations. This paper is a scoping review of existing literature to collate evidence on the performance of TB supply chains in the WHO African region. The WHO African region was chosen because it bears the second largest burden of TB in the world [9], yet in general its supply chains are considered to be structurally weaker relative to other regions [10]. We followed the scoping review approach developed by Arksey and O'Malley (2005) [11] because we aimed to summarize and disseminate research findings in relation the performance of TB supply chains in the WHO African region. According to Arksey and O'Malley [(2005)], that kind of scoping study might describe in more detail the findings and range of research in particular areas of study, thereby providing a mechanism for summarizing and disseminating research findings to policy makers, practitioners and consumers who might otherwise lack time or resources to undertake such work themselves. This aligned with our aim of informing stakeholders on the performance of TB supply chains to guide future strategies for enhancing performance.

The overall question for the scoping review was: What are the strengths and gaps in relation to TB supply chain performance in the WHO African region and what can be done to improve performance? We sought to answer the following specific questions: 1)] what are the strengths in relation to TB supply chain performance? 2) what are the limitations in relation to TB supply chain performance? and 3)] What recommendations are put forward to improve TB supply chain performance?

The rest of the paper is organized as follows. First, we introduce the conceptual aspects of supply chain management, with an inclination towards TB. We then present a framework for evaluating the performance of supply chain for TB commodities in the context of logistics functions. That would be followed by an overview of TB PSM in the context of disease related aspirations and declarative commitments. We then present the methods followed by the results. The results are then discussed and conclusions drawn.

### Supply chain management for TB commodities: Conceptual aspects

Supply chain management (SCM) encompasses all the activities aimed at planning, coordinating and managing the flow of products from the source to the end user, and the information

flow associated with it [12]. Although aspects related to the supply of commodities is often referred to as 'logistics', there exists a conceptual distinction between logistics and SCM. Theoretically, logistics is a very old term that originated in the military, for the maintenance, storage, and transportation of military personnel and property [13]. The primary concern for a logistics system is to fulfil the six "rights": ensuring that the right goods, in the right quantities, in the right condition, are delivered to the right place, at the right time, for the right cost [14]. In the context of TB, logistics focuses on delivering the relevant commodities to facilities or sites that offer TB services, and ensuring that there are no delays in delivering those commodities. On the other hand, supply chain management encompasses an integrating function with primary responsibility for linking major business functions and business processes within and across companies into a cohesive and high-performing business model [15]. Thus, for TB, this would involve ensuring that the relevant commodities are available on the market, and there are well coordinated efforts to bring the commodities to the users in a transparent, efficient and equitable fashion. Logistics is therefore but only one, yet essential component of supply chain management whose primary function is to facilitate the physical movement of TB products. Despite these conceptual distinctions, logistical functions in the health sector have been presented within the broader lens of supply chain management, and this has potentially helped to foster a continuous connection between the two. Since this paper is not aimed at refining the conceptual nuances related to the field of supply chain, where relevant the term supply chain management will be applied to models that are originally conceived as logistics. One widely accepted model in relation to health products is the logistics cycle from the USAID| DELIVER Project [16].

### The logistics cycle: Putting TB supply chain performance into perspective

The logistics cycle is depicted to consist of repetitive major activities namely product selection, quantification and procurement, inventory management and serving customers. To align with public health terminology, in this study we used the term service delivery instead of serving customers. The activities in the center of the logistics cycle represent the management support functions that inform and impact the other elements around the logistics cycle. Supply chains operate within a broader policy environment that has to be adaptable and quality monitoring should be embedded in all the processes and activities. The logistics cycle is presented below Fig 1.

Broadly, supply chain performance is concerned with the extent to which supply chain activities meet end-customer requirements, including product availability, on-time delivery, and all the necessary inventory and capacity in the supply chain to deliver that performance in a responsive manner [17]. When analyzing system performance, qualitative evaluations such as "good", "fair", "adequate", and "poor" are vague and difficult to utilize in any meaningful way. As a result, quantitative performance measures are often preferred to such qualitative evaluations [18]. The performance of public health supply chains is of heightened relevance for various reasons. From the economic perspective of the principal-agent theory [19], the end users of health products or patents (principals rely on the agent (health personnel) to choose the most appropriate products for them according to their specific needs and conditions. Therefore, unlike other products, health products are not easily substitutable. This renders public health supply chains particularly vulnerable to market failures i.e., the failure of supply chain to ensure availability of a specific health product might mean forgone care altogether or a matter of 'life and death'. For example, in the context of TB, the unavailability of a medicine for treating drug resistant TB would mean forgone treatment for an eligible patient, even if the same facility is overstocked with medicines that treat other forms of TB. These market failures impose high performance expectations on public health supply chains.

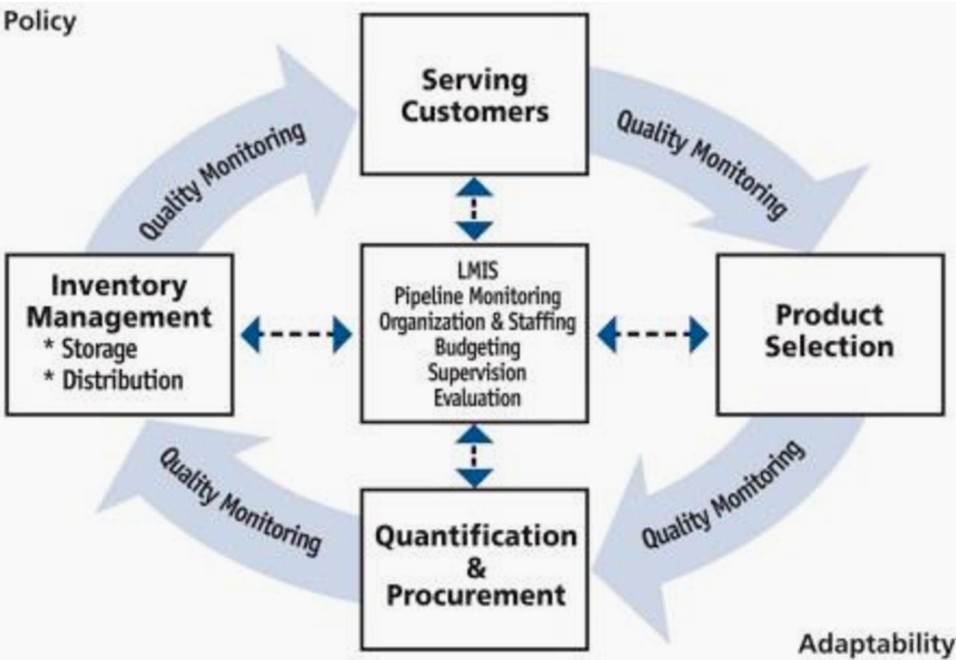

**Fig 1. The logistics cycle.** Source: USAID|DELIVER PROJECT.

The performance of commodity supply chains also influences the public perception on health programs. Arguably, medicines are among the most tangible representation of the performance of entire health systems, and their availability enhances public confidence in the credibility the entire health system [20]. The performance of public health supply chains is also of huge interest from an economic or financial stand point, including amongst external funders. For example, every year, roughly half of the Global Fund's investments–about US$2 billion–is used to procure health commodities and strengthen PSM systems [21]. The performance of PSM systems is therefore fundamental to the overall performance of programs supported by external financiers. As mentioned earlier, supply chains remain a linchpin for attaining a number of TB related commitments. Table 1 below places supply chain within the context of various strategies and commitments.

## Methods

We conducted a qualitative research based on a scoping review of available literature related to TB supply chain performance.

### Search strategy

We followed the scoping review approach developed by Arksey and O'Malley (2005) [11] and following the PRISMA Extension for Scoping Reviews (S1 Checklist). Between October and November 2023, we conducted a literature search in three electronic databases PubMed/MEDLINE, Global Index Medicus and Academic complete. We used the search terms: *tuberculosis AND supply chain management AND (developing countries or developing nations or third world or low income countries), tuberculosis AND medicines AND drugs AND stock outs AND (developing countries or developing nations or third world or low income countries)*. We also searched Google scholar and google search engines.

**Table 1. Supply chain in the context of TB related strategies, declarations and commitments.**

| STRATEGES | | | |
|---|---|---|---|
| **Strategy** | **Main provisions** | **Framing of supply chain** | **Role of effective supply chin performance** |
| Directly-observed treatment, short-course [1994] | Essential basics to address the TB epidemic [political will, diagnosis, treatment and monitoring] | Establishment of a system for regular drug supply of all essential anti-TB drugs | Ensure availability of essential diagnostics and medicines |
| STOP TB Strategy [2006–2015] | DOTS expansion and enhancement, addressing the emerging challenges of HIV-associated TB and MDR-TB, access to quality TB care through multi-sectoral engagement and research | Revitalized emphasis on reliable supply chain systems as part of DOTS | Ensure availability of essential diagnostics and medicines with new emphasis on addressing upstream determinants of supply e.g., market availability |
| The End TB Strategy *[2015]* | Patient centred care, policy and systems, research and innovation | No explicit mention of supply chain systems | Introduction of new vaccines, drugs and diagnostics |
| **DECLARATIONS / COMMITMENTS** | | | |
| **Declaration/commitment** | **Aspiration /context** | **Framing of supply chain** | **Role of effective supply chin performance** |
| Amsterdam Declaration to Stop TB [2000] | Expansion of DOTS and 70% detection of infectious cases by the year 2005 | IMPROVING systems of procurement & distribution of tuberculosis drugs | Ensure availability of essential diagnostics and medicines |
| The End TB Strategy [2015] | Reduction in number of TB deaths by 90% by 2030 compared with 2015 | No explicit mention of supply chain system but an emphasis on access to new vaccines, drugs and diagnostics | Reliable supply chain systems key for the introduction of new tools |
| Moscow Declaration to End TB [2017] | Detection of at least 90 per cent of cases and successful treatment of at least 90 per cent of those detected | No explicit mention of supply chain system but an emphasis on access to new drugs and diagnostics | Reliable supply chain systems key for the introduction of new tools |
| United Nations High level Meeting on TB [2018] | Treat 40 million people with TB between 2018 and 2022, 3.5 million children with TB, 1.5 million people with drug-resistant TB and at-least 30 million put on TB Preventive Treatment | No explicit mention of supply chain system but an emphasis on access to new and optimized drugs and diagnostics | Reliable supply chain systems key for the introduction of new tools |
| The Political Declaration of the UN High-Level Meeting on the fight against tuberculosis [2023] | treatment for up to 45 million people between 2023 and 2027, including up to 4.5 million children and up to 1.5 million people with drug-resistant tuberculosis and provision of TPT to 45 million people | No explicit mention of supply chain system but an emphasis on access to new and optimized drugs and diagnostics | Reliable supply chain systems key for the introduction of new tools |

## Study selection

To be eligible, a publication had to meet all of the following criteria: 1)conducted in the WHO African region 2)tackle any aspect related to TB supply chain performance 3) describe strengths and gaps in relation to the performance of TB supply chain. No time limit was set for the search.

## Data analysis

### Data charting

We created a data extraction tool in Microsoft Excel capturing how the contents of each study included from the scoping review aligned with the elements of supply chain performance across the domains of the logistics cycle. The tool also captured how supply chain performance is framed within those studies, including linkages with the programmatic processes (e. g. timely diagnosis for diagnostics) and outcomes (e.g. treatment success rates for medicines). One author extracted the verbatim excerpts from the documents and posted them into the respective areas through own interpretive analysis. After the initial categorization, the material was re-read and the large text verbatim excerpts were progressively reduced to find common patterns and themes from the documents. Other three authors independently reviewed the themes and provided feedback.

## Results

### Identification of studies

We retrieved 1, 564 publications from our initial database searches and 22 from google scholar and google searches. Form these, 26 duplications were removed. After removing duplicates, the records were screened for the relevance of the title and abstract, which led to the exclusion of 1,487 records and retention of 63 records that we sought for retrieval. Out of these, we managed to retrieve all of them and these were further assessing for full text eligibility. Of these, 38 were excluded for the following reasons: Tackling supply chain not related to TB (n = 17), TB supply chain performance aspects not clearly spelt out(n = 14)and insufficient details on TB supply chain performance (n = 7). Overall, 25 studies were included in this review. Fig 2 below shows the PRISMA diagram flow for the identification of the studies.

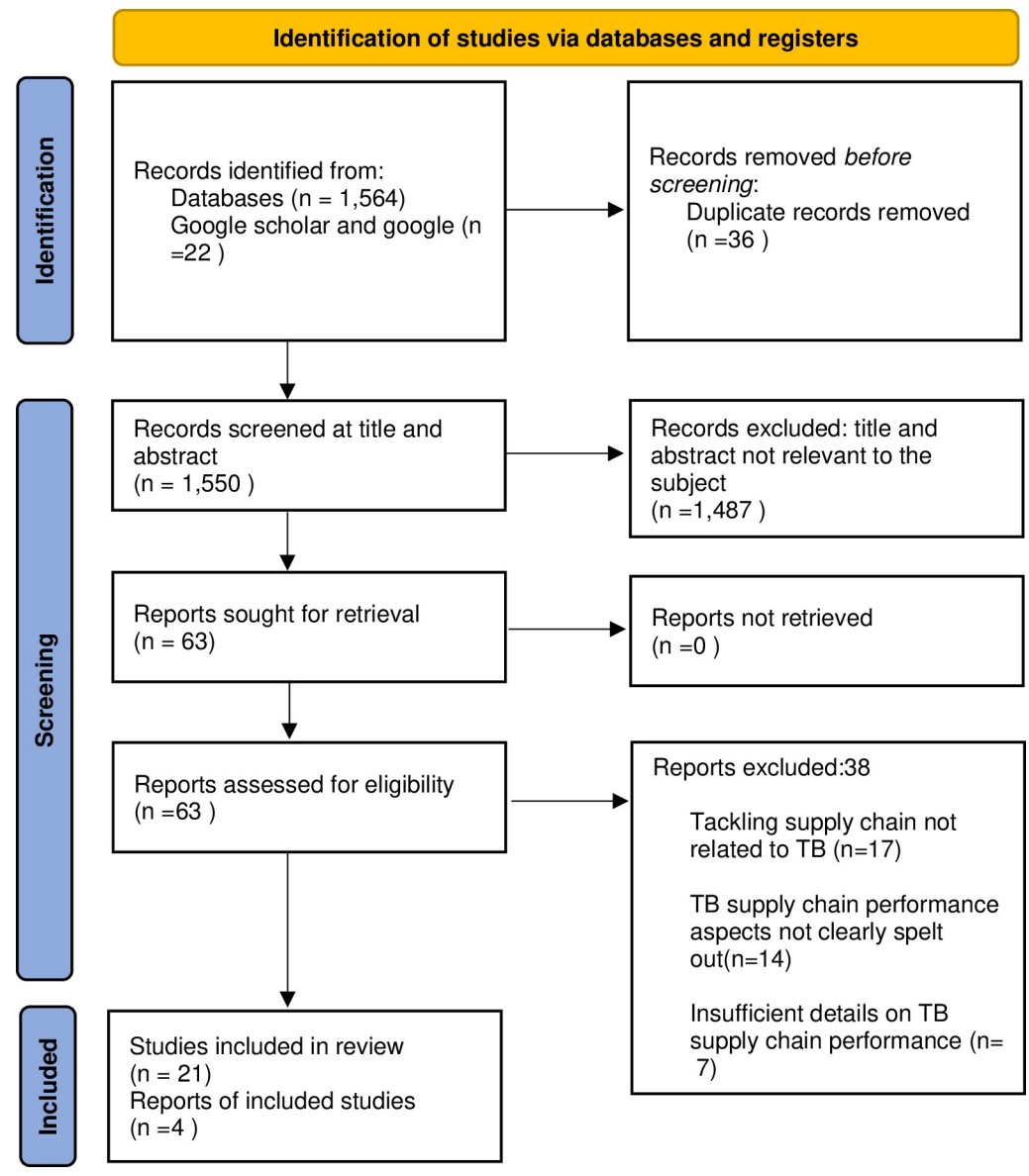

**Fig 2. PRISMA flow diagram.**

**Table 2. Study characteristics.**

| Main performance metric | Number of studies (%) | References |
|---|---|---|
| Availability | 5 (20%)] | [22–26] |
| Availability and distribution | 1 (4%) | [27] |
| General* | 9 (36%) | [28–35] |
| Logistics Management Information Systems (LMIS) | 4 (16%) | [36–39] |
| Service delivery** | 6 (24%) | [40–45] |

*General means the study tackled various stages of the logistics systems without a dominant focus on one area
*Service delivery means the study had a major focus on the impact of supply chain performance on patient management at the point of care (facility)

Twenty-five (25) studies were included in the full review. Out of these, 21 are papers published in peer reviewed academic journals whilst four (4) are reports. The studies were conducted from 2009 to 2023. Twenty-four (24) studies focused on a single country and one was multi-country. Overall, the studies covered 12 countries (Botswana, Ethiopia, Lesotho, Liberia, Malawi, Nigeria, South Africa, Tanzania, Uganda, Zambia and Zimbabwe) Out of these studies, four countries contributed nearly 70% namely Ethiopia (28%), Uganda (16%), Nigeria (12%) and South Africa (12%). Table 2 below shows study characteristics according to the stage of the logistics cycle.

## Summary of studies reviewed

Table 3 below shows the summary of studies.

## What are the strengths in relation to TB supply chain performance?

Our scoping review identified a number of strengths in relation to TB supply chain performance. *Availability*: In relation to availability of commodities, some countries had no stock outs [23, 41], or experienced minimum stock outs [43].

**Inventory management.** One study found that majority of health facilities used health commodity management information system [30]. Data flow among the central, regional and the district levels on total quantities of stock on hand and total numbers of patients on treatment by regimen was found to be available in one study [46]. Others studies found high availability of recording and reporting tools, high completion rates and on- time reporting [23, 39]. Four countries reported that they have set the required minimum and maximum stock levels and have defined ordering and sock taking schedules [34]. Average stock accuracy rate was found to be high in one study [30] whilst use of digitized LMIS was associated with reduced stock-outs [38].

**Quantification and procurement.** One study identified that quantification at the national level was done by using the forms provided by the Global Drug Facility (GDF) based on the morbidity method (i.e., the number of patients needing treatment for the quantification period) [32]. In relation to supplier performance, one study found high fulfillment of key performance indicators by suppliers [24]. Another study found that ordered quantities matched the annual forecast while all of the orders placed through GDF during that year were delivered in full and on time during the period reviewed [23]. A multi-country study showed that all countries had in place PSM plans that complied with the external funders, such as the Global Fund [34].

**Table 3. Summary of studies reviewed.**

| Author | Title | Year | Article type | Study country | Commodity | Performance metric |
|---|---|---|---|---|---|---|
| USAID DELIVER | Uganda national tuberculosis logistics system assessment tool (LSAT) | 2009 | Report | Uganda | Medicines | General |
| Bello SI | Challenges of DOTS implementation strategy in the treatment of tuberculosis in a tertiary health institution, Ilorin, Nigeria | 2010 | Peer reviewed journal publication | Nigeria | Medicines and diagnostics | General |
| Chana RC | Malawi: Assessment of the Logistics and Supply Chain Management of Anti-TB Medicines | 2011 | Report | Malawi | Medicines | General |
| USAID DELIVER | Tanzania: TB and Leprosy Logistics System Assessment | 2011 | Report | Tanzania | Medicines | General |
| Desale A | Assessment of laboratory logistics management information system practice for HIV/AIDS and tuberculosis laboratory commodities in selected public health facilities in Addis Ababa, Ethiopia | 2013 | Peer reviewed journal publication | Ethiopia | Diagnostics | LMIS |
| Cattamanchi A | Health worker perspectives on barriers to delivery of routine tuberculosis diagnostic evaluation services in Uganda: a qualitative study to guide clinic-based interventions | 2014 | Peer reviewed journal publication | Uganda | Diagnostics | General |
| Seunandeni T | The extent and impact of TB drug stock-outs | 2014 | Peer reviewed journal publication | South Africa | Medicines | Availability |
| Jatau B | Procurement and Supply Management System for MDR-TB in Nigeria: Are the Early Warning Targets for Drug Stock Outs and Over Stock of Drugs Being Achieved? | 2014 | Peer reviewed journal publication | Nigeria | Medicines | Availability |
| Mavere T | Rapid Situation Analysis of the Five East, Central, and Southern Africa Countries on TB Data and Commodity Management | 2014 | Report | Malawi, Swaziland, Tanzania, Uganda, and Zambia | Medicines | General |
| Sinishaw MA | Distribution and Availability of Essential Tuberculosis Diagnostic Items in Amhara Region, Ethiopia | 2015 | Peer reviewed journal publication | Ethiopia | Diagnostics | Distribution and availability |
| Sinishaw MA | Longer lead time of tuberculosis laboratory commodities in Amhara region, Ethiopia | 2016 | Peer reviewed journal publication | Ethiopia | Diagnostics | LMIS |
| Chamusingarevi I | Evaluation of A Tuberculosis Control Programme | 2016 | Peer reviewed journal publication | Zimbabwe | Medicines and diagnostics | General |
| Stillson CH | 'That's when I struggle'. . . Exploring challenges faced by care givers of children with tuberculosis in Botswana | 2016 | Peer reviewed journal publication | Botswana | Medicines | Serving customers |
| Bam L | Reducing stock-outs of essential tuberculosis medicines: a system dynamics modelling approach to supply chain management | 2017 | Peer reviewed journal publication | South Africa | Medicines | Availability |
| Tadesse L | Evaluation of Quality of Tuberculosis Care in Limmu Genet District Hospital, Oromia Region, Ethiopia | 2017 | Peer reviewed journal publication | Ethiopia | Medicines and diagnostics | Serving customers |
| Scott NA | Optimizing drug inventory management with a web-based information system: The TBTC Study 31/ACTG A5349 experience | 2017 | Peer reviewed journal publication | Multiple | Medicines | LMIS |
| Tiye K | Logistics management information system performance for program drugs in public health facilities of East Wollega Zone, Oromia regional state, Ethiopia | 2018 | Peer reviewed journal publication | Ethiopia | Medicines | LMIS |
| Desta K T | Performance of the National Tuberculosis Control Program in the post conflict Liberia | 2018 | Peer reviewed journal publication | Liberia | Medicines and diagnostics | General |

(*Continued*)

**Table 3.** (Continued)

| Author | Title | Year | Article type | Study country | Commodity | Performance metric |
|---|---|---|---|---|---|---|
| Koomen LEM | Effects and determinants of tuberculosis drug stock outs in South Africa | 2019 | Peer reviewed journal publication | South Africa | Medicines | Availability |
| Kalema N | Gaps in TB preventive therapy for persons initiating antiretroviral therapy in Uganda: an explanatory sequential cascade analysis | 2020 | Peer reviewed journal publication | Uganda | Medicines | Serving customers |
| Oluwasan MM | General and tuberculosis-specific service readiness in two states in Nigeria | 2020 | Peer reviewed journal publication | Nigeria | Medicines and diagnostics | Serving customers |
| Tola FB | Anti-Tuberculosis Commodities Management Performance and Factors Affecting It at Public Health Facilities in Dire Dawa City Administration, Ethiopia | 2020 | Peer reviewed journal publication | Ethiopia | Medicines and diagnostics | General |
| Nidoi A | Impact of socio-economic factors on Tuberculosis treatment outcomes in northeastern Uganda: a mixed methods study | 2021 | Peer reviewed journal publication | Uganda | Medicines and diagnostics | Serving customers |
| Sintayehu K | Determinants of stock-outs of first line anti-tuberculosis drugs: the case of public health facilities of Addis Ababa city administration health bureau, Addis Ababa, Ethiopia | 2022 | Peer reviewed journal publication | Ethiopia | Medicines | Availability |
| Andom AT | Understanding barriers to tuberculosis diagnosis and treatment completion in a low resource setting: A mixed-methods study in the Kingdom of Lesotho | 2023 | Peer reviewed journal publication | Lesotho | Medicines and diagnostics | Serving customers |

## What are the gaps in relation to TB supply chain performance?

**Stock outs.** Commodity availability or its inverse (stock-outs) was the most measured metric in relation to TB supply chain performance. The studies showed that stock outs are a recurrent problem in a number of countries. Studies showed that medicines for treating TB in children were stocked out for the last six months [30] and one year [26]. Another study showed that about a fifth of facilities surveyed had stock-outs of any ARV or TB drug [22]. Shortages of anti-TB drugs were considered the major problem in TB treatment and care in one study [29] whilst stock outs of medicines used for preventing TB were found to be prevalent in two studies study [40, 42]. Stock outs also extended to laboratory commodities. Stock outs of key diagnostics was reported three studies [27, 29, 44]. In relation to the programmatic impact, one study reported that stock out led to service interruption for a duration of up to 15 days [37].

**Weak logistics management information systems and inventory management.** In one study, health personnel did not have adequate knowledge on the use of electronic LMIS [30]. Another study reported that the laboratory logistics management information system was weak and consistently being hampered by poor communication[37]. With respect to inventory management, studies also reported low stock accuracy, delayed reporting and incomplete and/or inaccurate reports [34, 36, 39]. One study found poor adherence to maximum, minimum and buffer stock levels; with little knowledge and understanding of these parameters across facilities [32] whilst another study reported that over than 70% of treatment centers placed an order when stock was below the minimum level [23]. Another study found that the maximum-minimum–maximum inventory levels were not set up accurately which induced the risk of stock outs [46].

**Differential effects of stock outs.** The studies also found differential effects of stock outs. In South Africa, one study reported that stock out proportions were higher and TB treatment outcomes were seen to be worse in districts with higher poverty [25]. The studies also showed

that some product categories are more affected than others. The extent of stock out for TPT medicines appeared to be severe [40], a critical barrier for the implementation and scale up of the intervention [42] and a source of desperation and hopelessness amongst caregivers [40]. In relation to medicines used for treatment, formulations for children were found to be more vulnerable to stock outs compared to adult formulations [26, 30, 45].

Stock outs had severe adverse impacts for program performance. One study showed that a 10% rise in TB drug stock out proportions resulted in a 2.14% decline of the TB cure rate and a 1.43% decline of the TB treatment success with a positive correlation between stock outs and higher death rates [25]. The negative consequences were largest in the poorer districts where a 10% increase in TB drug stock outs led to a 3.25% decline in the TB cure rate and a 2.78% decline in the TB success rate [25]. Stock outs were found to be associated with financial hardship as patients incurred out-of-pocket expenditures or for go care altogether [31, 44]. Stock outs of medicines for children led to higher rates of loss to follow up amongst children [45].

**Weak ordering and distribution systems.**   Delayed distribution was found to be a major challenge in one study [23]. Delayed supply of medicines by central medical stores was found to significantly increase facility-level stock out of at least one anti-TB medicines by 10 times [26]. A more centralized system was found to increase transportation delay by about four folds as compared to health center trucks [37]. The proportion of drug orders received at the treatment centers in full and on time was found to be very low in one study [23]. Sub-optimal order fill rates were also found to be a major cause of stock outs. In one study, stock out of at least one anti-TB drug from health facilities significantly increased by 11 times when the central medical stores reduced the quantities ordered [26].

**Weak quantification, procurement and financing systems.**   One study reported weak quantification, largely due to the low skills and lack of capacity staff was a major driver of stock outs [22]. In relation to procurement, one study revealed longer lead times due to low responsiveness by suppliers [24]. In another study, there was poor adherence to procurement schedule leading to delays in delivery and uncoordinated shipments that led to stock outs and expiries [35]. Limited registered suppliers for childhood formulations was also found to be a major determinant of stock outs in two studies [34, 35]. Another study reported inefficiencies due to fragmentation of the supply chain and market uncertainties [24]. The financing of TB commodities was also found to have major gaps. Lack of a dedicated budget for procurement and delays in the disbursement of funds was found to be major challenge in two studies that examined countries that use domestic resources for procurements [34, 35].

## What are the recommendations for improving the performance of TB supply chain?

This scoping review identified a range of strategies and interventions that can be put in place to address deficiencies in TB supply chain performance. We also identified some approaches and practices that are associated with enhanced TB supply chain performance. Regarding quantification, timely and periodic forecasting and updating of supply plans was found to reduce stock outs and expiries [35]. In addition, the use of standardized quantification tools that are based on validated approaches, such as those provided by GDF was found to be effective in reducing over or under quantification [32]. Adherence to supply plans was also found to be a strong determinant of effective supply chain performance [23]. In relation to procurement, the capacity to identify suitable suppliers and monitor their performance was identified as key in enhancing the performance of TB supply chains. Preferentially selecting drug suppliers with short lead times was associated with the efficiency of the entire downstream supply chain [24]. In line with this, one study recommended strong linkages between procurements,

shipment, and funding considerations to anticipated commodity needs of the system [35]. Availability of funds and its timely release by the Global Fund [donor] may have contributed to effective procurement in one study [23].

The studies put forward a number of recommendations aimed at improving in-country performance of TB supply chains. In terms of system design, the recommended maximum-minimum inventory levels should be expertly designed in consideration of the overall length of the pipeline (the time required for a commodity to travel through each level of storage and distribution before reaching the client). To mitigate against expiries, the overall pipeline should not be longer than the shelf-life of medicines [46]. Regarding inventory management, the set minimum and maximum levels need to be standardized and adhered to at the various tiers of the health system [32]. The other system design parameter relates to buffer or safety stock. To enhance agility (defined as the ability to withstand unplanned changes in external variables without causing a shortage), the system should have in-built safety stock [24]. With regard to ordering, studies recommended a careful consideration whether commodities can be distributed based on facilities determining the quantities they require (pull system)or a central unit determining the quantities (push system). The urging principle of the central medical stores i.e., forced ordering system or no report no drug principle was associated with consistent and timely reporting of logistics data [39]. For effective coordination, there is need for designated staff responsible for PSM within the national TB control programs [46].

## Discussion

This is one of the few studies to collate evidence on the performance of TB supply chains through a scoping review methodology. The study identified a number of strengths and gaps in relation to TB supply chain and related recommendations to enhance performance. The main strength is in relation to the existence of relevant global and in-country structures and systems to support TB supply chains. At the global level, the existence of pooled procurement mechanisms such as the GDF was found to have a positive effect on TB supply chain performance. Although the role of GDF is not elaborated in many studies, two studies highlighted the organization's role in the critical role of the organization in the provision of standardized tools for quantification and capacity building [32] and coordination of procurements [23, 32]. The relative desirable performance of GDF supported countries aligns with the benefits associated with other Pooled Procurement Mechanism, including for products such as for HIV/AIDS and Malaria [47]. In relation to financing, Global Fund support was found to have a positive effect on commodity availability [23]. In particular, Global Fund support was found to be particularly performance- enhancing in instances where funding was sufficient and aligned with program needs alongside efficiencies during budget execution [23]. This echoes other studies that emphasize the need for robust forecasting, predictable financing and effective budget utilization to enhance the performance of supply chains supported through Global Fund grants [48, 49].

Several strengths also exist in relation to in-country logistics. First, TB commodities are integrated within general supply chain systems including for platforms such LMIS. This is a desirable design feature, considering that integrated PSM systems are more effective and responsive [50] and offer value for money [51]. Second, TB commodities are decentralized and integrated within the primary health care system which enhances geographical access to diagnosis, prevention and treatment. This aligns with tenets of DOTS and the aspiration of ending TB by 2030 within the UHC discourse [52].

The study illuminates a number of challenges. At upstream level, countries that utilize domestic channels of procurement lack the purchasing power associated with pooled

procurement mechanisms and encounter a number of inefficiencies that hamper TB supply chain performance. This is consistent with the observation that countries [especially small countries] that procure independently are vulnerable to high or variable prices for the small volumes they purchase, encounter, limited product selection, lack of quality assured products [53]. The most pertinent challenge relates to stock outs. Despite the existence of structures and mechanism to ensure product availability, stock outs remain a critical challenge across several countries. In congruence with other studies, causes of stock outs include manufacturing shortages, errors in ordering or requesting products, weak distribution systems, unanticipated spike in demand, inadequate funding and staffing, and lack of incentives to maintain stocks [54]. The other challenge relates to non-compliance with existing systems, such as inventory management practices and LMIS. This aligns with observations from other studies which showed that the major constraint to effective supply chain performance is the sub-optimal or non-use of existing systems.

## Policy implications

This study raises a number of policy implications for global and national actors. For global actors, this study has shown supply chain performance has an important influence on overall on the fulfillment of a number of declarative commitments. This underscores that such global aspirations should explicitly frame PSM as a strategic issue as opposed to a supportive or 'logistical' function whose main role is confined to the physical flow of health products. The study points to several strategic directions that are worth examining at global level. One key aspect is in relation to the strategic initiatives aimed at leveraging PSM systems to address the inherent market failures for TB which are characterized by fragmented demand and weak purchasing power amongst the countries most at need [55]. To address these market failures, long term disease aspirations should be accompanied by market shaping efforts that are informed by global demand forecasting and consolidation. Therefore, whilst concerns regarding national ownership and sustainability may favor use of domestic channels of procurement [53], country level actors should consider use of centralized platforms such as pooled procurement mechanisms to leverage efficiencies.

Global actors should also examine the contextual realities of translating global commitments into national action. This study has shown that countries face various challenges in the design and execution of TB supply chain systems, with persistent stock out as a recurrent problem. Therefore, whilst disease related aspirations are often accompanied with global initiatives to enhance access to products, such as price reductions, it should not be assumed that country level supply chains will be ready handle such products. This is of particular relevance in light of the evidence that there is discordance between global framing of access to medicines and domestic prioritization of interventions [56]. It is therefore critical to examine how country level supply chains will handle products in commitment-laden environments, including health system readiness for launch and scale up of relevant interventions. To synergize global efforts and country level implementation, national actors should put in place transformational supply chain plans that align with such changes, whilst global actors can support such plans where there is need.

This study has also raised equity-related implications of supply chain performance that are cross cutting to global and national actors, including civic society organizations. Whereas stock outs are predominantly framed as a generalized problem, studies in our scoping review found that they disproportionately affect vulnerable populations, including children and other socio-economically disadvantaged groups. Relatedly, due to their positive correlation with out of pocket expenditure, stock outs are likely to induce financial hardship on the vulnerable

**Table 4. TB supply chain performance in the context of eliminating TB by 2030.**

| TB PSM performance area | Overview of main challenges in relation to eliminating TB by 2030 | Example of threat to elimination of TB by 2030 | Interventions to improve performance |
|---|---|---|---|
| Quantification | • Non- use of standardized tools<br>• Over and under -quantification<br>• Lack of quality data<br>• Inconsistent quantification updates | • Under quantification has resulted in severe stock outs of MDR-TB medicines, leading to high death rate | • Use of validated quantification tools including Early warning systems |
| Procurement | • Reliance on single or few suppliers<br>• Weak demand<br>• Inefficient domestic procurement systems | • Insufficient funds and poorly executed domestic budgets have led to stock outs of first line medicines | • Utilization of pooled procurement mechanisms including through domestic funds |
| In-country logistics | • Orders misaligned with needs<br>• Late ordering<br>• Poor order fill rates | • Inconsistent availability of medicines identified as a barrier to TPT uptake | • Data quality trainings<br>• Use of electronic LMIS systems<br>• Ensure full supply at central level |

segments of society, a direct contradiction to the ethos of the end TB strategy. For example, the end TB strategy envisages zero catastrophic costs on the part of individuals who seek TB care [52], a target that is unlikely to be met if stock outs continue to follow a social gradient. Considering that TB is inherently 'a disease of the poor', of note is that the pattern of stock outs tend to follow the 'inverse care law'; a situation where the availability of good medical care tends to vary inversely with the need of the population served [57]. Thus, it is important to frame stock outs within the discourse of health equity and their adverse effect on the achievement of the end TB strategy thereof. This aligns with some nascent school of thoughts that have proposed an equity-based solution to the problem of stock-outs in developing countries, including through measurements such as the stock-out severity index [58]. An equity focused supply chain agenda also aligns with the tenets of people centered supply chain as espoused by leading agencies that support public health supply chain globally [59, 60]. Table 4 below summarizes TB supply chain performance in the context of eliminating TB by 2030.

## Areas of further research

As a one of the pioneering scoping reviews in the area of TB supply chain performance, this study does not only provide potential policy implications, but also opens avenues for further research. Despite that the performance of TB supply chain is inherently influenced by the health system environment, there is dearth of studies that adopt a health systems approach. There is need for further studies that examine the influence of the health system on TB supply chain performance. Although this study focused on TB supply chain, it may have practical usefulness on other programs, including overall supply chain systems. There is therefore a need for a PSM research agenda that is more comprehensive and cross cutting, including the contribution of PSM to broader policy aspirations such as health systems strengthening (HSS) and universal health coverage (UHC). An area of particular interest is research on the equity dimensions of poor supply chain performance. Such a research agenda is timely, if not overdue, in light of the observation that supply chain remains one of the least studied area across health system pillars [61]. This might be attributed to the erroneous notion that PSM is 'operational' or limited knowledge on available methods and approaches for PSM research. However, this study has shown that established methods of health system research, such as scoping review, can be applied to PSM. Such approaches should be promoted to expand the knowledge base and improve the quality of evidence used to inform strategies for supply chain improvements. Regarding study methodologies, there is need to expand from survey based methods to case studies. This will assist in improving the state of knowledge since case study methodology

for PSM is associated with theory building, detailed explanations of "best practices" and understanding of data gathered [62].

## Limitations

The study has some limitations worth highlighting. First, the corpus of studies on TB supply chain performance is generally low which constrains effective pooling of knowledge in the area. However, our scoping review found an encouraging trend of PSM publications from countries such as Ethiopia, and other countries could learn from the country's experience regarding TB supply chain studies. Second, all the studies reviewed are cross-sectional and some date as back as pre-2015 which may not accurately depict current TB supply chain performance. Nevertheless, the study has highlighted various strengths and challenges, including some potential legacy issues that might be relevant in contemporary times. The other limitation is that most reviewed studies are descriptive in nature. This leads to vague generalizations of PSM performance such as poor 'logistics' followed by imprecise and non-actionable recommendations for improving supply chain performance. There is need for more focused studies that are diagnostic in nature to examine the root causes of poor supply chain performance to improve the quality of recommendations. In this study, we found that studies that utilized the Logistics System Assessment Tool (LSAT)] [35, 46] developed by USAID|DELIVER PROJECT were more diagnostic in nature, and such comprehensive approaches should be promoted. The other limitation is that the study solely relied on publicly available literature. Such an approach might have left out relevant country specific reports that are not publicly available. An opportunity exists to use such sources for further studies. The third limitation is in relation to the geographical reach. By limiting the scope to the WHO African region, articles from other regions might have been left, which compromises the breadth and generalizability of the results.

## Conclusion

Pooled procurement mechanisms for TB commodities exist at global level to enhance efficiencies whilst countries have in place supply chain designs that promote availability and related information flows. However, poor supply chain performance remains a formidable threat to the overall program performance and achievement of End TB strategy. Stock outs of TB commodities are still prevalent, particularly for medicines used to treat TB in children and preventive therapy. Countries that rely on domestic resources and mechanism to procure TB commodities tend to be more vulnerable to stock outs due to inadequate financing, delays in fund disbursements, longer procurement lead times and poor supplier management. We recommend individual countries to utilize existing pooled procurement mechanisms that have inherent comparative advantages, including the flexibilities that exist to use domestic resources to procure commodities through those channels. Poor supply chain performance disproportionately affects already disadvantaged populations which calls for the design of supply chains that are sensitive to the differentiated needs of the people affected by TB. Overall, studies on TB supply chain are few. A research agenda is required to increase the scope of knowledge, including for supply chain systems in general.

## Supporting information

**S1 Checklist. PRISMA checklist.**
(PDF)

## Author Contributions

**Conceptualization:** Alison T. Mhazo.

**Data curation:** Alison T. Mhazo.

**Formal analysis:** Alison T. Mhazo, Stanford Miyango, Lifton Palani, Charles C. Maponga.

**Investigation:** Alison T. Mhazo.

**Methodology:** Alison T. Mhazo.

**Project administration:** Alison T. Mhazo.

**Supervision:** Alison T. Mhazo.

**Validation:** Alison T. Mhazo.

**Writing – original draft:** Alison T. Mhazo, Stanford Miyango, Lifton Palani, Charles C. Maponga.

**Writing – review & editing:** Alison T. Mhazo, Stanford Miyango, Lifton Palani, Charles C. Maponga.

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
