## [Decision Letter · Decision Letter 0]

26 Mar 2024

PGPH-D-24-00319

Tuberculosis commodities supply chain performance in the WHO African region: a scoping review

Dear Dr. Mhazo,

Thank you for submitting your manuscript to PLOS Global Public Health. After careful consideration, we feel that it has merit but does not fully meet PLOS Global Public Health’s publication criteria as it currently stands. Therefore, we invite you to submit a revised version of the manuscript that addresses the points raised during the review process.

The authors have written the article well and have put forward recommendations. It would be good to project the limitations of the study also. There are some missing points noted  by the reviewers also and correcting of those would increase the impact of the manuscript

We look forward to receiving your revised manuscript.

Kind regards,

Suma Thankamma Krishnasastry, MBBS, MD,DNB

Academic Editor

Journal Requirements:

2. Please provide separate figure files in .tif or .eps format only and remove any figures embedded in your manuscript file. Please also ensure all files are under our size limit of 10MB.

Additional Editor Comments (if provided):

Reviewers' comments:

Reviewer's Responses to Questions

**Comments to the Author**

1. Does this manuscript meet PLOS Global Public Health’s publication criteria? Is the manuscript technically sound, and do the data support the conclusions? The manuscript must describe methodologically and ethically rigorous research with conclusions that are appropriately drawn based on the data presented.

Reviewer #1: Yes

Reviewer #2: Yes

2. Has the statistical analysis been performed appropriately and rigorously?

Reviewer #1: N/A

Reviewer #2: N/A

3. Have the authors made all data underlying the findings in their manuscript fully available (please refer to the Data Availability Statement at the start of the manuscript PDF file)?

Reviewer #1: Yes

Reviewer #2: Yes

4. Is the manuscript presented in an intelligible fashion and written in standard English?

Reviewer #1: Yes

Reviewer #2: Yes

5. Review Comments to the Author

Reviewer #1: The authors addressed the main research question of summarizing the strengths and weaknesses of the existing TB supply chain mechanism in the WHO Africa region. They have put forth relevant recommendations like functional global mechanisms which utilized at country level to mitigate shortfalls in current supply chain mechanism. The manuscript is generally well written.

No major issues were identified however a few minor issues are noted below that need to be addressed:

1. The logistics cycle diagram should be labelled as a figure.

2. Table 1 -in the strategy column the “The political declaration of the UN High Level Meeting on the fight against tuberculosis” is not dated as the other strategies in the table on page 10

3. The authors mentioned four journal databases were used for the literature search however only three are name on page 12. The fourth database should be added.

4. The footnotes related to asterisk within the PRISMA flow diagram are missing on page 15

Recommendation- manuscript is well written overall and should be published.

Reviewer #2: 1. The research question was clearly outlined, well written and comprehensive to follow

2. Research strategy was well noted, multiple databases for literature search were used, appropriate search terms were utilized, and alternative sources were also considered.

3. Clear Inclusion and Exclusion Criteria (no questions)

4. Transparent Reporting: The review was reported transparently, following the recommended the scoping reviewing approach.

5. The authors effectively summarize the findings of the study and connect them to broader implications for TB supply chain management.

6. Interpretation of results: The authors interpret their findings in the context of existing literature and global health initiatives related to TB control. They provide insightful explanations for the identified strengths and challenges, such as the impact of pooled procurement mechanisms and the persistent issue of stock outs.

7. Critical analysis: The discussion critically evaluates the implications of the study findings, particularly in relation to the effectiveness of current supply chain systems and the need for strategic interventions to address existing challenges.

Comments to consider:

1. DOT(S) is a bit obsolete terminology in the TB program, as apart from the direct observation strategy other modes of administration are used (family-, community-, self -administration).

2. Some good points mentioned in the part of the "Areas of further research" which needed to be highlighted and be considered in the Limitations.

6. PLOS authors have the option to publish the peer review history of their article (what does this mean?). If published, this will include your full peer review and any attached files.

**Do you want your identity to be public for this peer review?** For information about this choice, including consent withdrawal, please see our Privacy Policy.

Reviewer #1: No

Reviewer #2: No

---

## [Editor Report · Decision Letter 1]

10 Apr 2024

PGPH-D-24-00319R1

Tuberculosis commodities supply chain performance in the WHO African region: a scoping review

Dear Dr. Mhazo,

Thank you for submitting your manuscript to PLOS Global Public Health. After careful consideration, we feel that it has merit but does not fully meet PLOS Global Public Health’s publication criteria as it currently stands. Therefore, we invite you to submit a revised version of the manuscript that addresses the points raised during the review process.

The manuscrpt is generally well written by the authors. This satisfies the publication criteria of PGPH journal

Minor revision of the manuscrip is necessary for cetain corrections and clarifications so as to increase the impact

We look forward to receiving your revised manuscript.

Kind regards,

Suma Krishnasastry, MBBS, MD,DNB

Academic Editor

Journal Requirements:

2. We have noticed that you have uploaded Supporting Information files, but you have not included a list of legends. Please add a full list of legends for your Supporting Information files after the references list.
---

## [Editor Report · Decision Letter 2]

22 Apr 2024

Tuberculosis commodities supply chain performance in the WHO African region: a scoping review

PGPH-D-24-00319R2

Dear Mr Mhazo,

We are pleased to inform you that your manuscript 'Tuberculosis commodities supply chain performance in the WHO African region: a scoping review' has been provisionally accepted for publication in PLOS Global Public Health.

Best regards,

Suma Krishnasastry, MBBS, MD,DNB

Academic Editor